# Longitudinal Changes in Serum Creatinine Levels and Urinary Biomarkers in Late Preterm Infants during the First Postnatal Week: Association with Acute Kidney Injury and Treatment with Aminoglycoside

**DOI:** 10.3390/children8100896

**Published:** 2021-10-09

**Authors:** Sang-Yoon Lee, Jung-Eun Moon, Sook-Hyun Park

**Affiliations:** 1Department of Pediatrics, School of Medicine, Kyunpook National University, Daegu 41404, Korea; gguggusy@gmail.com (S.-Y.L.); subuya@hanmail.net (J.-E.M.); 2Department of Pediatrics, Kyungpook National University Chilgok Hospital, Daegu 41404, Korea

**Keywords:** acute kidney injury, aminoglycoside, urinary biomarkers, serum creatinine

## Abstract

We aimed to determine the incidence of acute kidney injury (AKI) and longitudinal changes in SCr levels and urinary biomarkers associated with AKI and aminoglycoside (AG) medication during the first week of life of late preterm infants. Urine biomarkers and SCr were measured in thirty late preterm infants on days one, two, five, and seven postnatal. Urine biomarkers included neutrophil gelatinase-associated lipocalin (NGAL), monocyte chemotactic protein-1 (MCP-1), epidermal growth factor (EGF), Tamm–Horsfall glycoprotein (THP), and liver fatty-acid-binding protein (L-FABP). Gestational age was positively correlated with SCr levels at birth, but inversely correlated with SCr levels at day five and day seven. Eighteen (60%) infants had stage 1 AKI, and twenty (67%) infants were treated with AGs. Infants with AKI had lower gestational age and lower birth weight than those without AKI. Urinary biomarkers adjusted according to uCr levels in infants with AKI were not statistically different from those in infants without AKI. There were no significant differences in incidence of AKI, and SCr levels during and after cessation of AG treatment. The uMCP-1/Cr ratio at days five and seven was higher in infants treated with AG than in non-treated infants.

## 1. Introduction

Acute kidney injury (AKI) is common in preterm infants, particularly in very-low-birth-weight infants, and is significantly associated with high mortality during the neonate period. In adults, there is a high risk of chronic kidney disease and hypertension [1,2,3]. The incidence of AKI in neonatal intensive care unit has been reported to range from 15% to 56% [4,5,6], and several clinical factors such as hemodynamic unstable conditions (sepsis and patent ductus arteriosus); nephrotoxic medication involving aminoglycosides (AG), diuretics, non-steroidal anti-inflammatory drugs, and inotropes; inadequate volume status; and hypoxic status are known to affect the development of AKI in preterm infants [7,8,9,10,11]. In utero, nephrogenesis is supposed to be completed by 36 weeks of gestation. Therefore, preterm infants are born in the process of active renal formation, which makes them much more vulnerable to potential risk factors of AKI than full-term infants, including the risk of developing abnormal nephrogenesis or losing nephrons after birth [12,13,14].

Most neonatal AKI develops within the first week after birth [15]. The definition and severity of neonatal AKI is assessed by measuring the degree of increase in serum creatinine (SCr) level, although SCr levels have several limitations. SCr levels show a significant delay in increase, based on the definition of AKI, after approximately 25–50% of renal function is lost [2,7], which makes SCr levels difficult to detect during the early phase of AKI [15]. Therefore, SCr is considered an indicator of renal function, not renal injury. It is also difficult to differentiate the causes of AKI [16,17]. SCr levels are affected by various factors such as gestational age, birth weight, sex, muscle mass, and intravasular fluid status. Particularly in newborns, maternal SCr levels may affect neonatal SCr level during the first several days of life [15,17,18]. 

Currently, many studies are being conducted on the association between urinary biomarkers and AKI in preterm infants to detect the early stages before changes in SCr levels occur and overcome the limitation of neonatal SCr levels. Several studies have reported that urine biomarkers detected and predicted AKI as well as mortality in neonates [1,2]. Thus, urinary biomarkers can be used as indicators for monitoring nephrotoxicity as well as the effects of medication and therapeutic intervention; predicting the recovery of renal function after AKI; and classifying the regions of the injured kidney [19]. Most previous studies on AKI biomarkers in newborns have focused on extremely preterm infants who needed fluid therapy during the early period of life. Since fluid intake can also influence SCr levels and urinary biomarkers, we investigated changes in SCr levels and urinary biomarkers in late preterm infants with immature kidney function and sufficient oral intake without fluid supplement during physiologic weight loss.

We therefore investigated: (1) longitudinal changes in SCr levels and urinary biomarkers during physiologic weight loss; (2) correlation between SCr and urinary biomarkers; (3) characteristics of changes in urinary biomarkers during exposure to AG; and (4) clinical factors and changes in urinary biomarkers of the AKI group in late preterm infants during the first week of life.

## 2. Materials and Methods

### 2.1. Study Population and Ethics

Late preterm infants who were delivered at Kyungpook National University Children’s Hospital between March 2016 and April 2017 were enrolled in this study. Thirty late preterm infants were included and their urine biomarkers and SCr levels on days one, two, five, and seven postnatal were measured. We excluded patients with any chromosomal or major congenital anomalies and infants who needed parenteral nutrition during the first week of life. The study was approved by the institutional review board of Kyungpook National University Chilgok Hospital (IRB No. 2016-01-007). Informed consents were obtained from the participants’ parents.

### 2.2. Maternal and Neonatal Demographic and Clinical Data

Maternal and neonatal demographic data were collected through a review of relevant medical records. The maternal demographic features that were recorded included maternal SCr levels, premature rupture of membranes (PROM), gestational diabetes, and pregnancy-induced hypertension (PIH). We also collected information on the maternal use of antibiotics and steroids within one week before delivery. Neonatal clinical data included gestational age, birth weight, sex, delivery mode, Apgar score at 1 min and 5 min, weight loss during the first postnatal week, use of positive ventilation, and history of oligohydramnios. We collected neonatal medication history, including the levels of AG, diuretics, steroids, ibuprofen, and inotropes, which are known as nephrotoxicity-inducing drugs. 

In our unit, infants who had suspected sepsis or pneumonia were treated with ampicillin and AG (gentamicin) as an empirical antibiotics therapy. The dose of gentamicin was 5 mg/kg/dose every 36 h. For infants treated with AG, the duration of medication was within five days, and the levels of SCr and urinary biomarkers were evaluated two days after cessation of antibiotics. 

### 2.3. Measurement of Serum Creatine Levels and AKI Biomarkers

SCr levels were analyzed by an i-STAT analyzer (Abbott, Chicago, IL, USA) using 95 μL of capillary blood sampled from infants’ heels. Urine creatinine (UCr) level was measured using the urease glutamate dehydrogenase method (mg/dL, AU 5800, Beckman Coulter, Brea, CA, USA). Urine samples were collected using a sterile container. Particulates were removed by centrifugation for 15 min at 1000× *g*, and the samples stored at −80 °C until use. ELISA was performed according to the manufacturer’s instructions. Microtiter plates pre-coated with a monoclonal antibody against human epidermal growth factor (EGF, DEG00) Immunoassay (ng/mL, R&D Systems, Minneapolis, MN, USA), monocyte chemoattractant protein-1 (MCP-1, DCP00) Immunoassay (pg/mL, R&D Systems, Minneapolis, MN, USA), neutrophil gelatinase-associated lipocalin (NGAL, DLCN20) Immunoassay (μg/mL, R&D Systems, Minneapolis, MN, USA), Tamm–Horsfall glycoprotein (THP) ELISA (CSB-E09451) kit (ng/mL, Cusabio Biotech Co., Houston, TX, USA), and Liver Fatty Acid Binding Protein (L-FABP) ELISA (MBS017865) kit (ng/mL, MyBiosource, San Diego, CA, USA), were added with 100 μL of urine samples or standards for 1 h at 37 °C. After removing the liquid, each well was incubated with 100 μL biotinylated monoclonal antibody for 1 h at 37 °C. The solution was collected and washed three times after 100 μL avidin-conjugated horseradish peroxidase treatments for 1 h at 37 °C. After aspiration and washing 5 times, 100 μL tetramethylbenzidine substrate was applied for color development in the dark room, which was read after 10–30 min at 450 nm with ELISA reader (Benchmark Plus, BioRad, Hercules, CA, USA). All measurements were performed in duplicate, and all urinary biomarkers were evaluated after adjusting by matching each uCr level (mg/dL) correctly for the different state of urine dilution. 

### 2.4. Definition of AKI

Neonatal AKI was determined based on the modified Kidney Disease Improving Global Outcomes (KDIGO) classification, which defined stage 1 as an increase in the SCr by at least 0.3 mg/dL within 48 h or 1.5–1.9 times baseline within seven days. Stage 2 and stage 3 were defined as an increase of 2.0–2.9 times and ≥ 3.0 times baseline or an increase in the SCr to ≥2.5 mg/dL, respectively [20]. We did not apply urine amount to stage AKI, but we measured urine amount during the first week of life. Oliguria was defined <0.5 mL/kg/h of calculated urine output. 

### 2.5. Statistical Analysis

Statistical analysis was performed using IBM SPSS Statistics 26.0 software (IBM Corp., Armonk, NY, USA). The Shapiro–Wilk test was applied to evaluate normal distribution. The SCr levels and urinary biomarker levels on days one, two, five, and seven postnatal were compared using the Kruskal–Wallis test with Tukey’s post-hoc test using ranks. The demographic and clinical data between groups were compared using Mann–Whitney U test and Fisher’s exact test for continuous and categorical variables, respectively. The correlation between the two groups was analyzed by Spearman correlation. Statistical significance was set at a *p* value of <0.05.

## 3. Results

### 3.1. Demographic and Clinical Features of the Study Population

Demographic and clinical characteristics of the patients are shown in Table 1. Median gestational age of the population was 34.2 (34.0–35.3) weeks, and median birth weight was 2130 (1960–2300) g. Median weight loss was −1.0 (−2.4–1.0) % of their birth weight during the first postnatal week. Eighteen (60%) infants were diagnosed with stage 1 AKI, and 20 (67%) infants were treated with AG. There were no infants with oliguria, and stage 2 or 3 AKI. No infant was treated with ibuprofen, diuretics, or inotropes. Maternal median SCr was 0.56 (0.49–0.66) mg/dL, and none of the mothers had a medical history of AKI or chronic kidney disease before and during pregnancy.

### 3.2. Comparison of Changes in Sr and Urinary Biomarkers during the First Week of Age 

Table 2 shows the longitudinal changes in SCr levels and urinary biomarkers for all participants. SCr levels at day two were significantly elevated compared with those at days one, five, and seven. The uEGF/Cr and uTHP/Cr ratios at day two decreased compared with that at day seven. Apart from these differences, there were no statistically significant increase or decrease in SCr levels and urinary biomarkers during the first postnatal week.

The uNGAL/Cr and uEGF/Cr ratios of female infants were higher than those of male infants during the first week of life. However, uMCP-1/Cr, uTHP/Cr, and uL-FABP/Cr ratio did not differ by sex.

### 3.3. Correlation between Serum Creatinine and Urinary Biomarkers

Maternal SCr levels were significantly correlated with neonatal SCr at birth (ρ = 0.627, *p* < 0.001), but not with SCr at days two, five, and seven (Figure 1). Gestational age was positively correlated with SCr levels at birth (ρ = 0.392, *p* = 0.035), but inversely correlated with SCr levels at days five (ρ = −0.484, *p* = 0.008) and seven (ρ = −0.461, *p* = 0.013). All urinary biomarkers, adjusted according to each uCr level, were not correlated with gestational age and birth weight. The uNAGL/Cr (ρ = −0.427, *p* = 0.033), uMCP/Cr (ρ = −0.635, *p* < 0.001) and uEGF/Cr (ρ = −0.444, *p* = 0.016) ratios at birth were correlated with SCr at day two. There was no significant correlation between urinary biomarkers and SCr at birth, day five, and day seven. 

### 3.4. Comparison of Clinical Characteristics and Urinary Biomarkers between Infants with and without AKI

Infants in the AKI group had lower gestational age [34.1 (34.0–34.3) vs. 35.1 (34.4–35.5) weeks, *p* = 0.016] and lower birth weight [1990 (1923–2190) vs. 2240 (2180–2450) g, *p* = 0.016] than those in the non-AKI group (Table 3). Infants with AKI presented with lower SCr levels at day one, but higher SCr levels at days five and seven than infants without AKI. There were no significant differences in clinical factors, including weight loss, during the first week, Apgar scores, positive pressor ventilator, AG treatment, and maternal medication history between the two groups. Infants with AKI of uMCP/Cr ratio at day seven was higher than infants without AKI, but other urinary biomarkers corrected by uCr levels in infants with AKI were not statistically different from infants without AKI (Figure 2).

### 3.5. Comparison of Clinical Characteristics and Urinary Biomarkers between Infants Treated with Amionglycoside and Non-Treated Infants

Table 4 shows comparison between infants with and without AG treatment. There were no significant differences in maternal characteristics, gestational age, birth weight, incidence of AKI, and SCr levels during AG treatment and after cessation of AG. uMCP-1/Cr ratio at days five and seven of AG-treated infants was higher than that of non-treated infants; however, other urinary biomarkers corrected according to uCr levels in infants with AG treatment were not statistically different compared with those in non-treated infants (Figure 3).

## 4. Discussion

In the present study, SCr levels at day two were elevated compared with those at days one, five, and seven. Maternal SCr levels correlated with neonatal SCr at birth, but not with neonatal SCr levels at days two, five, and seven. As gestational age was lower, SCr levels were lower at birth but higher at days five and seven. All urinary biomarkers adjusted to uCr levels were not correlated with gestational age and birth weight in the present study. Infants in the AKI group had lower gestational age and lower birth weight than infants in the non-AKI group. During AG treatment and after cessation of AG, uMCP-1/Cr ratio at days five and seven of AG-treated infants was higher than that of non-treated infants. 

It is known that during the early postnatal period, neonatal SCr levels are significantly influenced by maternal SCr levels and the change in neonatal SCr levels is very wide [15]. Neonatal SCr levels are also associated with other clinical factors, including dehydration or fluid overloading, medication, gestational age, birth weight, and muscle metabolism [21]. In this study, neonatal SCr levels at birth correlated with maternal SCr levels; however, thereafter, there was no significant correlation between maternal and neonatal SCr levels during the first week of life. In late preterm infants who did not require fluid therapy, SCr levels were lower at birth, but higher at days five and seven as the gestational age was younger. Birth weight did not correlate with SCr levels before and after adjusting for gestational age.

The current definition of neonatal AKI is still based on SCr levels and urine output (UOP) according to the KDIGO classification [20], although SCr levels and UOP have limitations in defining AKI which included delayed SCr increase after renal injury, inability as a diagnostic marker of AKI site, and dynamic changes in SCr levels by renal maturation in neonates [16]. In preterm infants, according to the association between renal maturation and SCr levels, there was a trial to define neonatal AKI by applying different cutoff values for SCr levels by gestational age [22]. Higher cutoff values of SCr levels in very preterm infants had higher specificity to predict outcome than KDIGO classification [22]. According to the main mechanism of inducing AKI, such as through renal tubular ischemia, many studies on changes in biological and molecular levels have detected early renal injury and differentiated the site of AKI in preterm infants [7,15,16,17,18].

Saeidi B et al. reported that urinary biomarkers are affected by gestational age, sex, and postnatal age [18]. They found that uNGAL/Cr was associated with gestational age, sex, and postnatal age, and that uEGF/Cr and uTHP/Cr correlated with postnatal age, but not with sex [18]. In the present study, none of the urinary biomarkers significantly correlated with gestational age. uEGF/Cr and uTHP/Cr ratios at day two were lower than those at day seven, but other urinary biomarkers did not significantly change by postnatal age. Female infants had higher value of uNGAL/Cr and uEGF/Cr ratios than male infants during the first week of life. Previous studies demonstrated that urine NGAL concentrations in female infants were higher than in male infants [23,24] and this sex difference reported in childhood group [25], although the cause is still under investigation. Kidney and urine EGF were sensitive to estradiol in a mouse model [26] and EGF levels were greater in female than in male mice [27].

Previous studies reported that uNGAL, uMCP, and uL-FABP are elevated during AKI, but that uEGF and uTHP decrease [28,29,30,31,32,33]. THP decreases in acute tubular injury, which suggests that THP protects from the response of inflammatory mediators [30]. NGAL is not only the most well-known biomarker for AKI in infants but also a diagnostic value of renal recovery [28,31]. uL-FABP is also elevated during tubular injury and could differentiate from prerenal AKI [32]. The role of EGF was reported in obstructive uropathy, which could help in the recovery from tubular injury [33]. Urinary biomarkers change approximately 24 h before the increase in SCr levels based on AKI definition [16]. In our study, SCr levels at day two were elevated compared with those at days one, five, and seven, and uNAGL/Cr, uMCP/Cr and uEGF/Cr ratios at birth correlated with SCr levels at day two. Previous studies have reported the peak SCr levels at about one to three postnatal days in preterm infants similar to our study [34,35,36]. This may be attributed to delayed creatinine clearance and immature tubular reabsorption of creatinine, compared to relatively low GFR at this time [36]. Infants with AKI presented with lower SCr levels at day one, but higher SCr levels at days five and seven than infants without AKI. However, urinary biomarkers corrected by uCr levels in infants with AKI were not statistically different compared with infants without AKI.

Over 80% of medications received were antibiotics. AKI associated with nephrotoxic medication occurred in 9% of very-low-birth-weight infants, and lower birth weight and more exposure to nephrotoxic medications were risk factors for AKI in preterm infants [37]. The development of nephrotoxicity depends on accumulated AGs in the proximal tubule epithelial cells (PTECs) of the renal cortex, and intracellular AGs can cause PTECs apoptosis or necrosis by various pathways [38]. The degree of renal maturation and the type of aminoglycoside used were important determinants of the effect of AGs on tubular function [39], which may indicate that preterm infants are at a higher risk of AG-induced AKI than full-term infants. In very early preterm infants, uNAGL significantly increased without the definite changes in SCr levels during gentamicin medication [7]. In this study, nNAGL/Cr ratio during and after AG treatment was not different from the non-treated group, but uMCP-1/Cr ratios at days five and seven when AG treatment was terminated and after termination were higher than those of non-treated infants. Previous studies have shown that MCP-1 is associated with renal ischemic or toxic injuries such as those occurring during cardiac surgery [19].

There are several limitations in our study. Our sample size was small, and it did not include infants diagnosed with stage 2 or 3 AKI and accompanied by oliguria. Compared with previous studies, the range of gestational age in our study was narrow. Therefore, there was a limit to the correlation between gestational age and urinary biomarkers. However, we included participants who did not need fluid therapy and adjusted all urinary biomarkers according to uCr levels, which could more clearly show the longitudinal changes in urinary biomarkers and SCr levels during physiologic weight loss, as well as a more significant association between aminoglycoside medication and urinary biomarkers. The present study reported longitudinal changes in SCr levels and various urinary biomarkers in late preterm infants at the time of completion of nephrogenesis associated with AKI and exposure to AG medication. Contrary to previous studies that showed maternal SCr levels can affect neonatal SCr levels during a significant period of early life, only SCr levels at birth correlated with maternal SCr during the first week of life in the present study. Although there is a need for further investigations to determine how long maternal SCr levels affect neonatal SCr levels, our study suggests that neonatal SCr levels during the early period of life is a function of infants’ own renal function.

## 5. Conclusions

In late preterm infants, developing AKI was associated with lower gestational age and lower birth weight. However, urinary biomarkers were not different between AKI and non-AKI infants. During AG treatment and after cessation of AG, there were no significant differences in SCr levels between AG-treated and non-treated infants, but uMCP-1/Cr ratios at days five and seven were higher than those of non-treated infants.

## Figures and Tables

**Figure 1 children-08-00896-f001:**
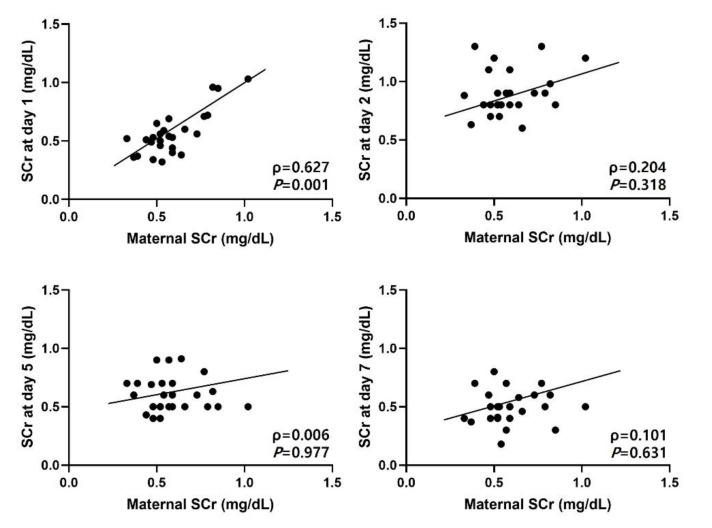
Correlation between maternal and neonatal serum creatinine levels. SCr, serum creatinine levels).

**Figure 2 children-08-00896-f002:**
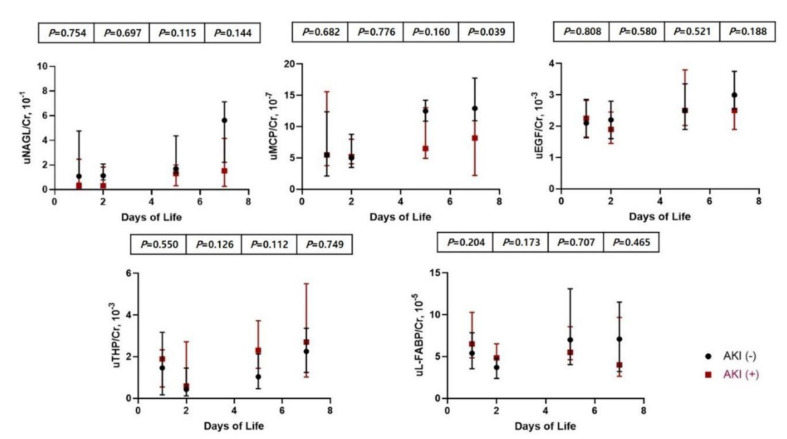
Comparison and changes in urinary biomarkers between infants with and without AKI. AKI, acute kidney injury.

**Figure 3 children-08-00896-f003:**
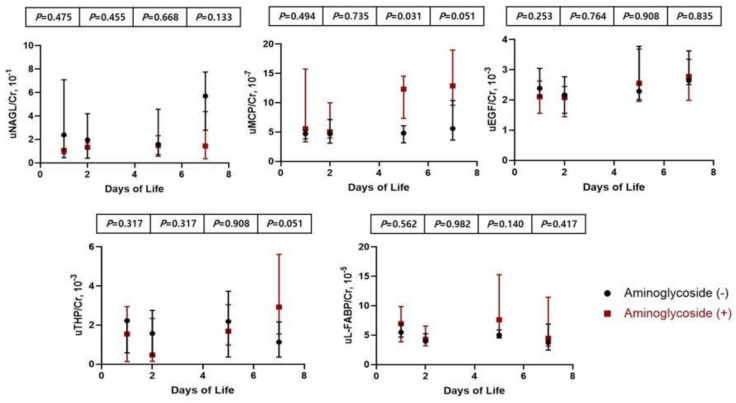
Comparison and changes in urinary biomarkers between infants with and without aminoglycoside treatment.

**Table 1 children-08-00896-t001:** Demographic and clinical characteristics of study group.

Variables	
Male, *n* (%)	12 (40)
Gestational age, wks	34.2 (34.0–35.3)
Birth weight, *g*	2130 (1960–2300)
Cesarean section	20 (67)
Apgar score	
1 min	7 (7–8)
5 min	9 (8–9)
Weight loss, %	−1.0 (−2.4–1.0)
AKI	18 (60)
Oliguria	0
PPV	20 (67)
Oligohydramnios	2 (6.7)
Medication history	
Aminoglycoside	20 (67)
Steroid	0
Ibuprofen	0
Diuretics	0
Inotrope	0
Maternal characteristics	
Diabetes	7 (23)
PIH	3 (10)
PROM	11 (37)
Steroid	9 (30)
Antibiotics	8 (27)
BUN, mg/dL	6.9 (5.9–9.6)
Creatinine, mg/dL	0.56 (0.49–0.66)

Data reported as frequency *n* (%) or median (IQR). AKI, acute kidney injury; PPV, positive pressure ventilator; PIH, pregnancy-induced hypertension; PROM, premature rupture of membrane; BUN, blood urea nitrogen.

**Table 2 children-08-00896-t002:** Longitudinal changes in serum creatinine levels and urinary biomarkers during the first postnatal week.

	1st	2nd	5th	7th	*p*-Value
Maternal SCr, mg/dL	0.56 (0.49–0.66)	0.56 (0.49–0.66)	0.56 (0.49–0.66)	0.56 (0.49–0.66)	-
SCr, mg/dL	0.53 (0.46–0.69)	0.88 (0.80–0.98)	0.60 (0.50–0.70)	0.50 (0.40–0.60)	<0.001
uNGAL/Cr, 10^−1^	1.09 (0.55–2.47)	1.34 (0.38–1.91)	1.49 (0.63–2.95)	2.21 (0.39–6.57)	0.497
uMCP-1/Cr, 10^−7^	5.47 (3.50–15.27)	5.01 (3.92–9.22)	9.18 (5.56–13.64)	10.60 (4.11–17.01)	0.116
uEGF/Cr, 10^−3^	2.19 (1.62–2.88)	2.13 (1.45–2.62)	2.49 (1.96–3.78)	2.73 (2.00–3.63)	0.042
uTHP/Cr, 10^−3^	1.69 (0.22–2.77)	0.54 (0.20–2.50)	1.76 (0.81–3.72)	2.39 (0.99–4.31)	0.023
uL-FABP/Cr, 10^−5^	6.07 (4.21–9.12)	4.07 (3.48–6.26)	5.89 (4.58–10.47)	4.10 (3.07–11.10)	0.122

Post-hoc Tukey’s test using ranks. Data reported as median (IQR). SCr, serum creatinine; uNGAL, urine neutrophil gelatinase-associated lipocalin; Cr, creatinine; uMCP-1, urine monocyte chemotactic protein-1; uEGF, urine epidermal growth factor; uTHP, urine T-H glycoprotein; uL-FABP, urine liver fatty-acid-binding protein.

**Table 3 children-08-00896-t003:** Comparison of clinical characteristics between infants with and without AKI.

	AKI (*n* = 18)	Non-AKI (*n* = 12)	*p*-Value
Male	7 (39)	5 (42)	0.514
Gestational age, weeks	34.1 (34.0–34.3)	35.1 (34.4–35.5)	0.016
Birth weight, *g*	1990 (1923–2190)	2240 (2180–2450)	0.016
Weight loss, %	−0.65 (−1.55–0.75)	−1.0 (−3.6–0)	0.521
Cesarean section	12 (67)	8 (67)	0.534
Apgar score at 1 min	7.0 (7.0–8.0)	7.5 (7.0–8.0)	0.588
Apgar score at 5 min	9 (8.0–9.0)	9 (9.0–9.0)	0.265
PPV, *n* (%)	13 (72)	8 (67)	0.659
Aminoglycoside, *n* (%)	12 (67)	8 (67)	0.592
SCr at day 1, mg/dL	0.50 (0.41–0.54)	0.72 (0.60–0.96)	0.001
SCr at day 2, mg/dL	0.89 (0.80–1.05)	0.80 (0.80–0.94)	0.492
SCr at day 5, mg/dL	0.65 (0.50–0.70)	0.50 (0.45–0.55)	0.014
SCr at day 7, mg/dL	0.50 (0.41–0.66)	0.37 (0.30–0.50)	0.007
Maternal Characteristics			
SCr, mg/dL	0.52 (0.48–0.59)	0.66 (0.54–0.82)	0.085
Diabetes, *n* (%)	4 (22)	3 (25)	0.547
PIH, *n* (%)	1 (6)	2 (17)	0.316
PROM, *n* (%)	8 (44)	3 (25)	0.449
Steroid, *n* (%)	7 (39)	2 (17)	0.412
Antibiotics, *n* (%)	6 (33)	2 (17)	0.671

Data reported as frequency *n* (%) or median (IQR). AKI, acute kidney injury; PPV, positive pres sure ventilator; SCr, serum creatinine; PIH, pregnancy-induced hypertension; PROM, premature rupture of membrane.

**Table 4 children-08-00896-t004:** Comparison of clinical characteristics between infants with and without aminoglycoside treatment.

	Aminoglycoside (*n* = 20)	Non-Aminoglycoside (*n* = 10)	*p*-Value
Male, *n* (%)	9 (45)	3 (30)	0.694
Gestational age, weeks	34.2 (34.1–35.2)	34.3 (34.0–35.3)	0.871
Birth weight, *g*	2115 (1960–2320)	2190 (1930–2230)	0.729
Weight loss, %	−1.0 (−2.4–1.0)	−1.0 (−2.4–−0.2)	0.501
C-sec, *n* (%)	15 (75)	5 (50%)	0.339
	Aminoglycoside (*n* = 20)	Non-aminoglycoside (*n* = 10)	*p*-value
Apgar score at 1 min	7 (7–8)	8 (7–8)	0.923
Apgar score at 5 min	9 (8–9)	9 (8–9)	0.885
Positive pressure ventilation, *n* (%)	17 (85)	4 (40)	0.067
AKI, *n* (%)	12 (60)	6 (60)	0.412
SCr at day 1, mg/dL	0.56 (0.46–0.72)	0.52 (0.46–0.54)	0.390
SCr at day 2, mg/dL	0.85 (0.80–0.99)	0.88 (0.80–0.90)	0.729
SCr at day 5, mg/dL	0.5 (0.49–0.61)	0.69 (0.50–0.70)	0.085
SCr at day 7, mg/dL	0.5 (0.4–0.55)	0.50 (0.40–0.60)	1.000
Maternal Characteristics			
SCr, mg/dL	0.57 (0.50–0.73)	0.54 (0.47–0.59)	0.458
GDM, *n* (%)	4 (20)	3 (30)	1.000
PIH, *n* (%)	2 (10)	1 (10)	1.000
PROM, *n* (%)	6 (30)	5 (50)	0.237
Steroid, *n* (%)	6 (30)	3 (30)	1.000
Antibiotics, *n* (%)	6 (30)	2 (20)	1.000

Data reported as frequency *n* (%) or median (IQR). AKI, acute kidney injury; PPV, positive pressure ven tilator; SCr, serum creatinine; PIH, pregnancy-induced hypertension; PROM, premature rupture of membrane.

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
