# Peer review of "Longitudinal Changes in Serum Creatinine Levels and Urinary Biomarkers in Late Preterm Infants during the First Postnatal Week: Association with Acute Kidney Injury and Treatment with Aminoglycoside"

_children, 2021, doi:10.3390/children8100896_

Round 1

Reviewer 1 Report

Line 124

„Longitudinal changes in urinary biomarker levels were compared using the Kruskal–Wallis test with Tukey’s post-hoc test using ranks“ – Did you mean Dunns statistic? I am not aware of KW having Tukey post hoc statistic. Please elaborate/correct.

Line 137 “none of the 136 mothers had a medical history of AKI before and during pregnancy.” – what about CKD?

Line 206 “all urinary biomarkers according to” – maybe adjusted to Ucrreatine levels?

In comparison of clinical characteristics and urinary biomarkers between infants with AKI and without AKI there is no urine output parameter. Staging is not mentioned – is it AKI 1st stage, 2nd or 3rd?

Only in the discussion section it is specified, that stages 2 and 3 were not included as well as oliguric patients. Please update methodology.

Regarding aminoglycoside cohorts – did you monitor serum levels of aminoglycosides? What SOP is followed in your institution regarding AG administration/dosing and monitoring?

There are many flaws in AKI definition based on creatinine increases, which you already outlined in your paper. In discussion, you highlight that “Infants with AKI presented with lower SCr levels at day 1, but higher SCr levels at days 5 and 7 than infants without AKI… urinary biomarkers corrected by uCr levels with AKI were not statistically different compared with infants without AKI” [Lines 250-252] which also suggest that creatinine does not have good specificity for neonatal AKI.

The creatinine at day 2 were largest. Could you please elaborate in the discussion section, what are the possible reasons? Perhaps day 2 is associated with increased demand of fluids and mother cannot catch up with milk production (since neonates did not receive intravenous fluids)?

Line 281 [Although there is a need for further investigations to determine how long maternal SCr levels affect neonatal SCr levels, our study suggests that neonatal SCr levels during early period of life is a value of infants’ own  renal function]

I believe the phrasing should be altered, because "early" is general term. Your study demonstrated that beyond 1st day of life there is little correlation left between maternal and neonatal creatinine. What are possible explanations for such a short period of neonatal/maternal creatinine correlation? Earlier study data showed creatinine correlation lasting up to 72 hours after birth.

20 reference contains and KDIGO update on CKD, not an AKI.  28 Reference is not in proper format. Please correct.

Author Response

 We would like to thank all reviewers for your valuable comments and suggestions which have immensely helped to improve the paper’s quality and readability. We have addressed all comments from the reviewer. The changes in the manuscript have been colored red. I really appreciate your review.

Reviewer 2 Report

In this article, the author  reported longitudinal changes in SCr levels and various urinary biomarkers in late preterm infants at the time of completion of nephrogenesis associated with AKI and exposure to AG medication.  It‘s interesting point that  only SCr levels at birth correlated with maternal SCr during the first week of life in the present study, which was contrary to  previous studies. However, the manuscript needs to be improved.  

Major points:

1.The usage and dosage of AG need to be clearly explained in the trial.

2.There are only 18 patients suffering from AKI. Why did 20 patients need AG treatment?

3.In the Table 2, the data of maternal SCr levels would need supplied, thus it's more visual to get the effect on the infant.

4.Can you explain why were SCr levels at day 2 elevated, but decline in next 5, 7days?

5.Can you explain why do the sex affect the uNGAL/Cr and uEGF/Cr ration?

6.I want to know why did the infant without AKI have higher  SCr levels at the day 1?

7.The conclusion of ‘’urinary biomarkers corrected according to in infants with AG treatment were not statistically different compared with those in non-treated infants‘’ may be misled by the small sample, because others project have prove the potential application of urinary biomarkers in their infant trial.

8.Some conclusion from the table or figure arenot consistent with the fact listed by the data, please check it.

Author Response

(The authors gave the same response as above.)

Round 2

Reviewer 2 Report

After the modifcation, the manuscript may be suitable for the journal.